# White Blood Cell and Platelet Counts Are Not Suitable as Biomarkers in the Differential Diagnostics of Dementia

**DOI:** 10.3390/brainsci12111424

**Published:** 2022-10-23

**Authors:** Sebastian Schröder, Johannes Heck, Adrian Groh, Helge Frieling, Stefan Bleich, Kai G. Kahl, Jacobus J. Bosch, Benjamin Krichevsky, Martin Schulze-Westhoff

**Affiliations:** 1Department of Psychiatry, Social Psychiatry and Psychotherapy, Hannover Medical School, D-30625 Hannover, Germany; 2Institute for Clinical Pharmacology, Hannover Medical School, D-30625 Hannover, Germany; 3Centre for Human Drug Research, 2333 CL Leiden, The Netherlands; 4Leiden University Medical Center, 2333 CL Leiden, The Netherlands; 5Institute for General Practice and Palliative Care, Hannover Medical School, D-30625 Hannover, Germany; 6Medical Service of the German Armed Forces, 24119 Kiel, Germany

**Keywords:** white blood cells, platelets, dementia, biomarker, diagnostics

## Abstract

Apart from Alzheimer’s disease (AD), no biomarkers for the differential diagnosis of dementia have been established to date. Inflammatory processes contribute to the pathogenesis of dementia subtypes, e.g., AD or frontotemporal dementia (FTD). In the context of cancer or cardiovascular diseases, white blood cell (WBC) populations and platelet counts, as well as C-reactive protein (CRP), have emerged as biomarkers. Their clinical relevance in dementia, however, is currently only insufficiently investigated. In the present study, hematological and inflammatory parameters were measured in the peripheral blood of 97 patients admitted to the gerontopsychiatric ward of Hannover Medical School, a university hospital in Germany, for dementia assessment. The study population comprised 20 non-demented, depressed patients (control group) and 77 demented patients who were assigned to five different groups based on their underlying dementia etiology: AD, *n* = 33; vascular dementia, *n* = 12; mixed dementia, *n* = 21; FTD, *n* = 5; and Korsakoff syndrome, *n* = 6. We observed neither statistically significant differences regarding total WBC populations, platelet counts, neutrophil-to-lymphocyte ratio and platelet-to-lymphocyte ratio, nor CRP levels between the control group and the five dementia groups. CRP levels tended to be higher in patients with Korsakoff syndrome than in the control group and in AD patients. Thus, CRP could possibly play a role in the differential diagnosis of dementia. This should be investigated further in future prospective studies with larger sample sizes. WBC and platelet counts, by contrast, do not appear to be suitable biomarkers in the differential diagnosis of dementia.

## 1. Introduction

Dementia poses huge challenges to healthcare systems worldwide. On the one hand, the increasing average age of the world population is estimated to lead to a doubling of the number of dementia cases over the next 20 years. On the other hand, essentially, only symptomatic drug therapies for the treatment of dementia are available to date [1,2]. The monoclonal antibody aducanumab, which was approved in the United States in 2021, is expected to lead to a reduction of the neurotoxic peptide amyloid-beta (Aβ) in Alzheimer’s disease (AD), thus pursuing a curative approach; however, its clinical effectiveness is currently the subject of debate [3,4].

In the management of patients afflicted with dementia, differentiation between dementia subtypes is crucial. For example, in mild to moderate AD, acetylcholinesterase inhibitors (e.g., rivastigmine or donepezil) can be administered with the aim of slowing disease progression, while for behavioral symptoms in the context of frontotemporal dementia (FTD), selective serotonin reuptake inhibitors (SSRIs) have been shown to exert positive effects [5,6]. In AD, cerebrospinal fluid analysis often reveals a characteristic constellation of elevated phospho-tau and decreased Aβ. For all other types of dementia, however, no diagnostic or therapeutic biomarkers have been clinically established to date [7]. Nevertheless, a wide variety of potential diagnostic and therapeutic markers are currently undergoing testing in patient-based studies. In addition, as blood-based biomarkers, beta-site amyloid precursor protein cleaving enzyme (BACE)1 in incipient dementias and neurofilament light (NF-L) as a general marker for neurodegeneration are currently widely studied [8]. Furthermore, protein-based studies and analyses of DNA methylation appear to have predictive diagnostic potential in various forms of dementia [9,10]. Moreover, machine learning-based techniques are increasingly emerging as a promising method for biomarker evaluation [11].

The contribution of inflammatory processes to neurodegeneration has been studied extensively in various forms of dementia. For a long time, the paradigm of the central nervous system as an immune-privileged organ persisted, but in recent years, it has been demonstrated that neurodegenerative processes lead to dysfunctions of the blood–brain barrier [12,13]. Especially in AD, evidence is accumulating that besides Aβ deposition and tau pathology, microglia also play a pivotal role in pathogenesis [14,15]. These cells proliferate in the area of Aβ deposits and contribute in a complement-mediated fashion to the loss of synapses, as well as to neurotoxicity via secretion of proinflammatory cytokines [16]. Inflammatory processes also contribute to the destruction of neurovascular structures in the context of vascular dementia (VD) [17]. In patients with FTD, alterations in individual immune cell populations and increased activation of microglia have been reported [18]. By contrast, knowledge about the role of inflammatory processes in mixed dementias (i.e., dementias with vascular and Alzheimer’s components) and in Korsakoff syndrome is limited.

Measuring white blood cell (WBC) counts, platelet counts, or C-reactive protein (CRP) levels in peripheral blood is a simple and inexpensive way to screen for inflammatory processes. A distinction should be made between cells of innate immunity such as neutrophils or macrophages, which play a decisive role in immediate reactions to infectious pathogens, and lymphocytes as important mediators of adaptive immunity [19]. In particular, the neutrophil-to-lymphocyte ratio (NLR) and the platelet-to-lymphocyte ratio (PLR) have been established as biomarkers in cancer or acute coronary syndrome [20,21]. Besides, an increased NLR has recently been demonstrated as a marker of incident AD in three large patient cohorts [22,23,24]. In general, however, the role of peripheral blood cells and inflammatory markers such as CRP in dementia has scarcely been studied, with most available studies being limited to investigations in patients with AD [25,26]. Previous analyses most readily identified IL-6 as a suitable marker to differentiate between dementia and depression [26].

We hypothesized that quantitative differences in WBC populations, platelet counts, and/or CRP levels might be present between patients suffering from different types of dementia. To elucidate this, we analyzed hematological/inflammatory markers in 97 patients who underwent dementia assessment on the gerontopsychiatric ward of a large German university hospital.

## 2. Methods

### 2.1. Ethics Approval

This study was approved by the Ethics Committee of Hannover Medical School (No. 10502_BO_K_2022) and adheres to the Declaration of Helsinki (1964) and its later amendments (current version from 2013).

### 2.2. Study Design and Eligibility Criteria

The present research was designed as a retrospective cohort study. From January 2015 to December 2021, 502 patients underwent dementia assessments at the gerontopsychiatric ward of the Department of Psychiatry, Social Psychiatry and Psychotherapy at Hannover Medical School. Assessments comprised neuropsychological testing via Mini-Mental Status Examination (MMSE) [21], neuroimaging (either as cranial magnetic resonance imaging (cMRI) or cranial computed tomography (cCT)), and clinical behavioral observation. Based on these examinations, the type of dementia (AD, VD, mixed dementia, FTD, or Korsakoff syndrome) was determined. Patients in whom the diagnostic tests did not reveal the presence of dementia but who were diagnosed with depression served as controls.

Besides a lack of written informed consent, incompleteness of diagnostic examinations, the presence of psychiatric diseases other than those listed above, and factors that might have influenced hematological parameters—such as delirium, systemic inflammatory disorders (e.g., infections, hematologic diseases, or autoimmune/autoinflammatory diseases), and treatment with immunomodulatory drugs (e.g., disease-modifying anti-rheumatic drugs or immunosuppressants)—were defined as exclusion criteria. Demographic characteristics—i.e., age, sex, and International Statistical Classification of Diseases and Related Health Problems 10th Revision (ICD-10) diagnoses—were retrieved from patient records.

Of the 502 patients who underwent dementia assessment during the study period, a total of 97 patients fulfilled the eligibility criteria and were enrolled in the study.

### 2.3. Collection and Analysis of Blood Samples

Blood samples were collected from the patients on the morning of the day of admission and were processed further without delay. S-Monovette^®^ K2 EDTA-Gel and S-Monovette^®^ Serum-Gel (Sarstedt AG & Co. KG, Nümbrecht, Germany) were used as collection tubes. CRP was quantified with an immunoturbidimetric assay (Roche Diagnostics, Mannheim, Germany) on a Cobas^®^ 6000 analyzer (Roche Diagnostics, Mannheim, Germany). WBC counts and platelet counts were determined with a Sysmex XN-10^TM^ Automated Hematology Analyzer (Sysmex GmbH, Norderstedt, Germany). Measurement of WBCs included differential WBC analysis for neutrophils, eosinophils, basophils, monocytes, and lymphocytes. NLRs and PLRs were calculated using neutrophil, lymphocyte, and platelet counts (NLR = neutrophil count/lymphocyte count; PLR = platelet count/lymphocyte count).

### 2.4. Statistical Analysis

Descriptive statistical techniques were used to summarize the data. Patient characteristics are presented as absolute and relative frequencies for categorical variables and as medians with interquartile ranges (IQRs) and variances for quantitative variables (due to non-Gaussian distribution). Differences between the study groups (i.e., one control group of non-demented, depressed patients, and five dementia groups (AD, VD, mixed dementia, FTD, and Korsakoff syndrome)) with respect to quantitative variables were analyzed with median tests for independent samples. Median tests as non-parametric tests were chosen because the data of the investigated parameters were not normally distributed (as revealed by inspection of histograms, Q–Q plots, and Shapiro–Wilk tests). In addition, median tests were used because of considerable variance across samples. *p* values (two-sided) < 0.05 were considered statistically significant. All statistical analyses were conducted with IBM^®^ SPSS^®^ Statistics 28 (Armonk, New York, NY, USA).

## 3. Results

### 3.1. Study Population

The study population comprised a total of 97 patients: 20 non-demented, depressed patients (control group) and 77 demented patients. Demented patients were assigned to five different groups based on their underlying dementia etiology: AD, n = 33; VD, n = 12; mixed dementia, n = 21; FTD, n = 5; and Korsakoff syndrome, n = 6. There were no major discrepancies between the six study groups (i.e., one control group and five dementia groups) with respect to age or sex (Table 1). As expected, MMSE results were higher in the control group compared to the dementia groups. Since MMSE results had been used to differentiate between demented and non-demented patients at enrollment (see Section 2, Methods), no inferential statistics were conducted with regard to MMSE results.

### 3.2. Neuroimaging

76.3% (74/97) of the study population received cMRI, while cCT was performed in 21.6% (21/97). No in-hospital neuroimaging was conducted in 2.1% (2/97), as it had already been completed in the ambulatory setting. All patients in the control group received neuroimaging (cMRI: 85.0% (17/20); cCT: 15.0% (3/20)), as compared to 97.4% (75/77) of demented patients (cMRI: 74.0% (57/77); cCT: 23.4% (18/77); no neuroimaging: 2.6% (2/77)).

### 3.3. Analysis of White Blood Cell Populations and Neutrophil-to-Lymphocyte Ratios

We investigated potential differences in terms of WBC populations and NLRs between the six study groups by utilization of median tests for independent samples. We did not detect statistically significant differences for the investigated parameters between the study groups: WBC counts, global *p* value = 0.641; neutrophil counts, global *p* value = 0.704; lymphocyte counts, global *p* value = 0.830; NLRs, global *p* value = 0.126; basophil counts, global *p* value = 0.775; eosinophil counts, global *p* value = 0.364; and monocyte counts, global *p* value = 0.861 (Table 2).

### 3.4. Examination of Platelet Counts and Platelet-to-Lymphocyte Ratios

In analogy to WBC populations, we investigated potential differences in terms of platelet counts and PLRs between the six study groups by utilization of median tests for independent samples. We did not observe statistically significant differences for the investigated parameters between the study groups: platelet counts, global *p* value = 0.409; PLRs, global *p* value = 0.735 (Table 2).

### 3.5. C-Reactive Protein Levels

Next, we analyzed potential differences in CRP levels between the study groups and observed a trend toward statistical significance (global *p* value = 0.072) (Table 2). Therefore, we conducted pairwise comparisons between the study groups and detected statistically significant differences between patients affected by Korsakoff syndrome and the control group (median CRP level (IQR) 7.65 mg/L (4.23–12.13 mg/L) vs. median CRP level (IQR) 1.15 mg/L (0.53–4.30 mg/L); unadjusted *p* value = 0.005), as well as between patients affected by Korsakoff syndrome and patients affected by AD (median CRP level (IQR) 7.65 mg/L (4.23–12.13 mg/L) vs. median CRP level (IQR) 1.30 mg/L (0.65–3.95 mg/L); unadjusted *p* value = 0.006). However, after correction for multiple testing via Bonferroni correction, the *p* values for these two pairwise comparisons were no longer statistically significant (*p* = 0.078 and *p* = 0.094, respectively).

## 4. Discussion

The present study sought to investigate the role of WBC populations, platelet counts, derived hematological parameters such as NLR and PLR, and CRP levels as potential biomarkers for the differential diagnosis of dementia. We did not find statistically significant differences with respect to WBC counts, neutrophil counts, lymphocyte counts, NLRs, basophil counts, eosinophil counts, monocyte counts, platelet counts, PLRs, or CRP levels between non-demented, depressed patients and five groups of demented patients (i.e., patients affected by AD, VD, mixed dementia, FTD, or Korsakoff syndrome). CRP levels showed a trend toward statistical significance, and significant differences were detected when we conducted pairwise comparisons between patients affected by Korsakoff syndrome and the control group, and between patients affected by Korsakoff syndrome and AD patients.

While the investigation of WBC populations and platelet counts in the context of dementias has revealed partly contradictory results, a pronounced role of neutrophils in the pathogenesis of AD has been repeatedly postulated [29,30]. Several studies have also demonstrated higher NLRs in patients with AD compared with healthy controls [31,32,33]. Furthermore, NLR could be reproduced several times as a marker of incident AD, whereas this was not the case in VD [22,23,24]. A study by Rembach and colleagues suggested a correlation between NLR and Aβ burden, but this result was no longer statistically significant after correction for multiple testing [33]. It has been discussed in the literature whether the NLR naturally increases during aging and is thus also increased in patients with dementia who are generally older than non-demented patients [34]. In any case, our study suggests that total WBC populations and platelet counts, as well as derived hematological parameters such as NLR and PLR, are not suitable for differential diagnoses between different forms of dementia.

The significance of inflammatory markers such as CRP, which can readily be determined in peripheral blood, has also been investigated repeatedly, especially in AD, with partly contradictory results [26,35,36]. While several studies showed significantly higher CRP levels in patients with AD than in healthy controls, a study by Nilsson et al. showed that patients with VD had significantly higher CRP levels than those with AD [37,38]. By contrast, the results of the present work suggested a trend toward higher CRP levels in patients with Korsakoff syndrome.

Limitations of the present study mainly arose from its small sample size; for example, we were only able to enroll five and six patients into the FTD group and the Korsakoff syndrome group, respectively. As we did not perform a formal sample size calculation, but rather opted for a convenience sample of consecutively enrolled patients, our study might have been underpowered to detect statistically significant differences between the study groups, e.g., with respect to CRP levels. In addition, the retrospective design of the study and the lack of a healthy control group should be taken into account as limiting factors. The latter is important to consider, since in patients with depressive disorders, inflammatory processes have been reported to contribute to pathogenesis [39], which makes the use of non-demented but depressed patients as control group in the present study debatable.

Despite statistically non-significant results, our study opens avenues for further research. While we focused on the analysis of total WBC populations, differential WBC counts and derived hematological parameters, interestingly, a flow cytometry study by D’Angelo et al. found that in contrast to total lymphocyte counts, the frequency of individual lymphocyte subpopulations within the total peripheral lymphocyte pool (in this study T cell subsets specifically) was significantly elevated in patients with AD, VD, and mixed dementia strongly suggesting that immunophenotyping of lymphocyte subsets could be further explored for differential diagnostic purposes [40]. In the present study, we detected a trend toward statistical significance when comparing CRP levels across study groups. More specifically, patients affected by Korsakoff syndrome displayed higher CRP levels compared to non-demented, depressed patients, as well as AD patients. Future studies investigating the role of CRP levels and cellular immunophenotyping in dementia should conduct a formal sample size calculation and would benefit from enrollment of larger numbers of patients per study group. Thus, CRP in conjunction with immunophenotyping of immune cell subsets might emerge as potential serological and cellular biomarkers in the differential diagnosis of dementia. In this context, the determination of CRP could be used in the future as a component of risk calculations for the development of dementia, in addition to the quantification of NF-L and BACE1 as well as epigenetic and protein-based methods.

## Figures and Tables

**Table 1 brainsci-12-01424-t001:** Characteristics of the study population.

	Control	Alzheimer’s Disease	Vascular Dementia	Mixed Dementia	Frontotemporal Dementia	Korsakoff Syndrome
	*n* = 20	*n* = 33	*n* = 12	*n* = 21	*n* = 5	*n* = 6
Median age (IQR)/(Variance)— Years	76 (65–79) (83.7)	76 (72–82) (74.6)	79 (70–83) (79.5)	79 (76–84) (42.3)	65 (59–82) (152.5)	69 (61–77) (69.9)
Female sex— n (%)	13 (65.0)	26 (78.8)	6 (50.0)	14 (66.7)	4 (80.0)	1 (16.7)
Male sex— n (%)	7 (35.0)	7 (21.2)	6 (50.0)	7 (33.3)	1 (20.0)	5 (83.3)
Median MMSE ^a^ (IQR)/(Variance)— points	28 (26–29) (2.1)	17 (9–22) (48.7)	16 (8–24) (61.4)	16 (13–24) (56.3)	15 (15–22) (22.8)	20 (14–24) (42.7)

^a^ MMSE scores range from 0 to 30 points, with higher scores indicating better cognitive functions. The following cut-off values for defining cognitive impairment have been suggested in the literature [27,28]: no cognitive impairment = 27–30 points; mild cognitive impairment = 23–26 points; and severe cognitive impairment (i.e., dementia) = 0–22 points. IQR denotes interquartile range, MMSE Mini–Mental State Examination.

**Table 2 brainsci-12-01424-t002:** White blood cell populations, platelet counts, and C-reactive protein levels in demented and non-demented patients.

	Control	Alzheimer’s Disease	Vascular Dementia	Mixed Dementia	Frontotemporal Dementia	Korsakoff Syndrome	*p* Value ^a^
	*n* = 20	*n* = 33	*n* = 12	*n* = 21	*n* = 5	*n* = 6	
Median white blood cell count (IQR)/(Variance) [×10^3^/µL]	6.60 (5.50–8.60) (6.1)	6.80 (5.80–8.15) (1.7)	7.65 (6.33–7.98) (4.5)	6.90 (5.60–9.20) (7.2)	6.20 (5.00–21.95) (198.8)	7.05 (4.18–8.23) (5.8)	0.641
Median neutrophil count (IQR)/(Variance) [×10^3^/µL]	4.59 (3.05–5.43) (5.1)	4.52 (3.23–5.76) (2.3)	4.50 (3.67–6.36) (5.0)	4.12 (3.72–5.98) 5.6)	3.89 (3.02–5.28) (2.0)	4.50 (2.61–5.60) (3.1)	0.704
Median lymphocyte count (IQR)/(Variance) [×10^3^/µL]	1.40 (1.05–1.99) (0.4)	1.60 (1.11–2.00) (0.3)	1.48 (0.91–1.77) (0.3)	1.22 (0.80–1.89) (0.5)	1.32 (1.29–10.32) (58.7)	1.43 (1.04–1.76) (0.2)	0.830
Median NLR (IQR)/(Variance)	3.22 (1.91–4.34) (4.8)	2.81 (2.05–4.36) (5.6)	3.34 (1.90–6.68) 6.6)	3.91 (2.53–5.54) (13.4)	2.07 (1.19–2.78) (1.0)	2.91 (2.41–3.39) (0.6)	0.126
Median basophil count (IQR)/(Variance) [×10^3^/µL]	0.04 (0.02–0.09) (0.0)	0.04 (0.02–0.05) (0.0)	0.04 (0.01–0.05) (0.0)	0.03 (0.02–0.06) (0.0)	0.02 (0.01–0.06) (0.0)	0.05 (0.04–0.09) (0.0)	0.775
Median eosinophil count (IQR)/(Variance) [×10^3^/µL]	0.10 (0.10–0.22) (0.1)	0.10 (0.07–0.18) (0.1)	0.10 (0.06–0.18) (0.1)	0.18 (0.07–0.30) (0.1)	0.12 (0.05–0.23) (0.1)	0.16 (0.11–0.59) (0.4)	0.364
Median monocyte count (IQR)/(Variance) [×10^3^/µL]	0.52 (0.47–0.78) (0.1)	0.52 (0.41–0.63) (0.0)	0.49 (0.39–0.71) (0.1)	0.57 (0.42–0.78) (0.0)	0.62 (0.41–1.05) (0.2)	0.67 (0.38–0.83) (0.1)	0.861
Median platelet count (IQR)/(Variance) [×10^3^/µL]	249.5 (204.0–274.8) (3024.1)	237.0 (201.5–282.0) (7132.4)	246.0 (195.5–311.8) (3307.4)	232.0 (204.5–278.0) (6265.4)	315.0 (202.0–381.5) (9682.8)	224.5 (198.0–338.3) 6261.6)	0.409
Median PLR (IQR)/(Variance)	167.6 (105.3–255.8) (6993.9)	150.0 (111.3–241.9) (12947.4)	155.3 (121.9– 328.5) (11377.3)	191.7 (125.1–298.1) (11557.8)	170.1 (71.1–252.8) (11993.2)	199.6 (121.4–232.6) (4243.0)	0.735
Median CRP (IQR)/(Variance)— mg/L	1.15 (0.53–4.30) (7.4)	1.30 (0.65–3.95) (52.8)	3.15 (1.15–12.83) (48.2)	3.10 (0.90–8.10) (185.9)	0.90 (0.70–10.25) (63.5)	7.65 (4.23–12.13) (42.8)	0.072

^a^ Global *p* values of median tests for independent samples. CRP denotes C-reactive protein, IQR interquartile range, NLR neutrophil-to-lymphocyte ratio, and PLR platelet-to-lymphocyte ratio.

## Data Availability

The data that support the findings of this study are available upon reasonable request from the corresponding author.

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
