# Peer review of "White Blood Cell and Platelet Counts Are Not Suitable as Biomarkers in the Differential Diagnostics of Dementia"

_brainsci, 2022, doi:10.3390/brainsci12111424_

Round 1

Reviewer 1 Report

Authors measured the only few blood biomarker indicators for possible differentiations of different dementia.

WBC, platelet counts, and CRP were already investigated and reported as non-significant indicators in AD.

Authors did not reference numerous reports on blood biomarkers for the diagnosis and differential diagnosis of various dementia. 

Authors need to reflect their data in discussions with blood biomarkers.

Since authors collected the precious samples, it would be great to measures latest blood biomarkers in the literature.

Author Response

Reviewer 1:

Authors measured the only few blood biomarker indicators for possible differentiations of different dementia.

We agree with the Reviewer that our assessment of blood biomarkers is not comprehensive. However, we would like to politely point out that we did investigate the sizable number of ten different hematological parameters.

WBC, platelet counts, and CRP were already investigated and reported as non-significant indicators in AD.

We agree with the Reviewer that white blood cell and platelet counts as well as CRP levels have been reported as non-significant indicators in AD in previous studies. However, our study went further than previous studies insofar as it was not restricted to global WBC counts but also examined white blood cell subpopulations such as neutrophil, lymphocyte, eosinophil, basophil, and monocyte counts. In addition, we assessed derived hematological parameters such as the neutrophil-to-lymphocyte ratio and the platelet-to-lymphocyte ratio. Moreover, our study was not limited to Alzheimer’s disease but also enrolled patients affected by other dementia etiologies, such as vascular dementia, mixed dementia, frontotemporal dementia, and Korsakoff syndrome.

Authors did not reference numerous reports on blood biomarkers for the diagnosis and differential diagnosis of various dementia.

It is true that our study did not cover the entirety of (bio)medical about blood biomarkers in the differential diagnostics of dementia. Unfortunately, in our article we had to restrict ourselves to citing articles that we considered most important for the context of our study. Our manuscript does not claim to cover or discuss the entire (bio)medical literature about blood biomarkers in dementia. Such an enterprise would clearly be beyond the scope of our study and would be more pertinent for a review article on this topic, which was not the goal of our current manuscript. However, we have added some references regarding the investigation of epigenetic and protein-based biomarkers in the context of dementia.

Authors need to reflect their data in discussions with blood biomarkers.

We thank the reviewer very much for this kind hint. Accordingly, in the discussion part, we added the reference to the possibility of determining CRP in addition to other blood-based biomarkers for the potential calculation of risk scores for the development of dementia.

Since authors collected the precious samples, it would be great to measures latest blood biomarkers in the literature.

We thank the Reviewer for this valuable suggestion. However, the patients in our study did not provide written informed consent that would allow us to examine other parameters/biomarkers than those presented in our article. Therefore, we do not think that it is ethically possible to analyze additional biomarkers within the scope of our study. Nevertheless, we believe that the Reviewer’s proposal is both thoughtful and inspiring, and should definitely guide the design of future studies on the topic.

Reviewer 2 Report

This manuscript conducted a straightforward but interesting study on assessing potential biomarkers for dementia. The authors collected blood samples from patients with dementia and depression (i.e., control), respectively. The dementia related samples were further divided into different subtypes. Then the authors performed analyses of these grouped blood samples to conclude that WBC populations and platelet counts are not effective biomarkers to differentiate dementia groups vs. non-dementia group.

The manuscript was well written in general. I have some additional comments & questions for the authors to address:

1. It would be reasonable to also report the variance for each group in comparison;

2. Depending on how the sample variances look like, we can then decide if the median tests used in this manuscript is the best option. Otherwise, I would be more than curious to know why the authors chose this test over other tests, e.g., the Kruskal–Wallis test?

3. Have you tried to compare all dementia groups combined (n = 77) vs. non-dementia group (n = 20)? I'm just curious about how that may come out.

4. Before you leave the WBC and platelet out of the game, have you thought about performing multivariate analysis where a linear combination of the blood sample measurements may be a good predictor for dementia? 

Author Response

Reviewer 2:

The manuscript was well written in general.

We would like to express our gratitude to the Reviewer for the generous appreciation of our manuscript.

  1. It would be reasonable to also report the variance for each group in comparison.

We thank the Reviewer for this valuable statistical consideration. In accordance with the Reviewer’s suggestion, we now report variances for all continuous variables in separate rows in Tables 1 and 2. As the Reviewer may see, the reported variances are quite large. In addition, inspection of histograms, box plots, and Q–Q plots for the hematological parameters did not suggest normally distributed values, which was further supported by Kolmogorov–Smirnov and Shapiro–Wilk tests. Therefore, we preferred to report medians with interquartile ranges instead of means with standard deviations/variances in our initial submission. However, since the Reviewer kindly asked for variances, we would like to also report variances in the revised version of our manuscript.

  1. Depending on how the sample variances look like, we can then decide if the median test used in this manuscript is the best option. Otherwise, I would be more than curious to know why the authors chose this test over other tests, e.g. the Kruskal–Wallis test?

The option of conducting Kruskal-Wallis-Tests has also been discussed, but due to large variance we initially preferred median tests. In brief, Kruskal–Wallis tests confirmed the results of the median tests, with one exception: The global P value of the Kruskal-Wallis test for the CRP level was significant (P = 0.025) since the pairwise comparison of the control group (i.e., non-demented, depressed patients) vs. patients affected by Korsakoff syndrome was statistically significant (P = 0.003), and remained statistically significant after adjustment for multiple testing (i.e., Bonferroni correction; adjusted P = 0.047). Also, the median test showed a trend towards statistical significance, so we preferred a more conservative approach. For all other investigated hematological parameters the median tests and Kruskal–Wallis tests delivered similar (i.e., non-significant) results.

  1. Have you tried to compare all dementia groups combined (n = 77) vs. non-dementia group (n = 20)? I’m just curious about how that may come out?

We thank the Reviewer for this thoughtful suggestion and—accordingly—compared all dementia groups combined (n = 77) vs. the control group (i.e., non-demented, depressed patients; n = 20) by utilization of Mann–Whitney U tests. We did not detect statistically significant differences for any hematological parameter between all dementia groups combined and the control group: CRP level (mg/L), 2.30 (IQR 0.80–6.45) vs. 1.15 (IQR 0.53–4.30), P = 0.065; white blood cell count (× 103/µL), 6.90 (IQR 5.80–8.15) vs. 6.60 (IQR 5.50–8.60), P = 0.772; platelet count (× 103/µL), 237.00 (IQR 205.00–298.00) vs. 249.50 (IQR 204.00–274.75), P = 0.972; lymphocyte count (× 103/µL), 1.42 (IQR 1.07–1.87) vs. 1.40 (IQR 1.05–1.99), P = 0.708; neutrophil count (× 103/µL), 4.40 (IQR 3.59–5.70) vs. 4.59 (IQR 3.05–5.43), P = 0.993; eosinophil count (× 103/µL), 0.12 (IQR 0.07–0.21) vs. 0.10 (IQR 0.10–0.22), P = 0.904; basophil count (× 103/µL), 0.04 (IQR 0.02–0.05) vs. 0.04 (IQR 0.02–0.09), P = 0.843; monocyte count (× 103/µL), 0.53 (IQR 0.41–0.72) vs. 0.52 (IQR 0.47–0.78), P = 0.865; neutrophil-to-lymphocyte ratio, 3.08 (IQR 2.21–4.65) vs. 3.22 (IQR 1.91–4.34), P = 0.817; and platelet-to-lymphocyte ratio, 165.10 (IQR 121.81–268.55) vs. 167.65 (IQR 105.34–255.82), P = 0.649.

  1. Before you leave the WBC and platelet out of the game, have you thought about performing multivariate analysis where a linear combination of the blood sample measurements may be a good predictor for dementia?

We think that the Reviewer makes an interesting point here, and in general we agree that the idea of a multivariate analysis is very intriguing. However, as in our study the individual hematological parameters did not differ significantly between the dementia groups and the control group (except for the CRP level when analyzed with the Kruskal–Wallis test), we do not think that a multivariate analysis would generate additional statistical benefit in this specific constellation. Besides, we are concerned that a multivariate analysis would be much harder to interpret for clinicians than the individual hematological parameters when examining patients for the presence/absence of dementia. Therefore, provided that the Reviewer agrees, we would prefer not to add a multivariate analysis to our study.
